# Improving Cross-view Object Geo-localization: A Dual Attention Approach with Cross-view Interaction and Multi-Scale Spatial Features

## Abstract

Cross-view object geo-localization has recently gained attention due to potential applications. Existing methods aim to capture spatial dependencies of query objects between different views through attention mechanisms to obtain spatial relationship feature maps, which are then used to predict object locations. Although promising, these approaches fail to effectively transfer information between views and do not further refine the spatial relationship feature maps. This results in the model erroneously focusing on irrelevant edge noise, thereby affecting localization performance. To address these limitations, we introduce a **Cross-view and Cross-attention Module (CVCAM)**, which performs multiple iterations of interaction between the two views, enabling continuous exchange and learning of contextual information about the query object from both perspectives. This facilitates a deeper understanding of cross-view relationships while suppressing the edge noise unrelated to the query object. Furthermore, we integrate a **Multi-head Spatial Attention Module (MHSAM)**, which employs convolutional kernels of various sizes to extract multi-scale spatial features from the feature maps containing implicit correspondences, further enhancing the feature representation of the query object. Additionally, given the scarcity of datasets for cross-view object geo-localization, we created a new dataset called **G2D** for the "Ground→Drone" localization task, enriching existing datasets and filling the gap in "Ground→Drone" localization task. Extensive experiments on the CVOGL and G2D datasets demonstrate that our proposed method achieves high localization accuracy, surpassing the current state-of-the-art.

## 1 Introduction

Cross-view object geo-localization involves locating the specific position of an object in a reference image based on the object's coordinates in a query image, as illustrated in Figure 1. This task has diverse applications, including robot navigation (McManus et al., 2014), 3D reconstruction (Middelberg et al., 2014), and autonomous driving (Häne et al., 2017). Unlike traditional object geo-localization (Krylov et al., 2018; Nassar et al., 2019; Chaabane et al., 2021), which deals with localizing objects under similar viewpoints, cross-view object geo-localization presents significant challenges due to drastic changes in viewpoints, scale variations, and partial occlusions between ground-view and aerial-view images. To address these challenges, Sun et al. (2023) introduced the CVOGL dataset and proposed a cross-view object localization model called DetGeo. DetGeo uses a dot product between globally averaged query image features and reference image features to direct the model attention to the spatial location of the query object. Although this approach establishes spatial correspondences between different views to some extent, it lacks sufficient feature interaction due to the simplicity of the computation. Additionally, the use of global average pooling discards a significant amount of spatial information, hindering the model's ability to effectively learn and associate contextual information about the query object across different views. As a result, the model may focus erroneously on edge noise similar to the query object, impairing localization performance.

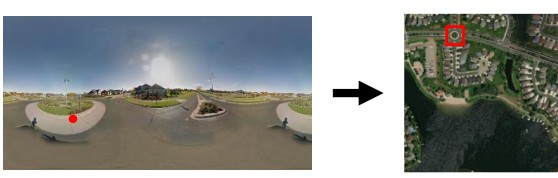 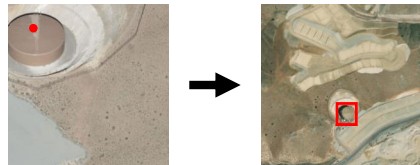

(a) Given a query object in the ground-view image (left), localize the query object in the satellite/drone-view image (right). The red dot indicates the query object, and the red box indicates the position of the query object in the satellite/drone-view image.

(b) Given a query object in the drone-view image (left), localize the query object in the satellite image (right). The red dot indicates the query object, and the red box indicates the position of the query object in the satellite image.

Figure 1: Cross-view object geo-localization task

To overcome these limitations, we draw inspiration from the core concepts of the Transformer (Vaswani et al., 2017) architecture and propose a Cross-view and Cross-attention Module (CVCAM). This module applies multiple iterations of cross-attention directly between the query image features and reference image features, promoting effective interaction between the two views. Additionally, it performs an extra cross-attention operation to integrate spatial information from both views. Such interaction and fusion operations of CVCAM enable the model to fully learn the contextual information of the query object across views, establish implicit correspondences, and suppress non-relevant edge noise. Furthermore, we propose a Multi-head Spatial Attention Module (MHSAM) to further process the spatial relationship feature maps output by CVCAM to enhance the representation of query object features. MHSAM consists of three attention heads that perform convolution and deconvolution operations with different kernel sizes, allowing the model to capture multi-scale spatial information from varying receptive fields. This approach significantly improves the model ability to capture local features of the query object while effectively reducing the representation of non-target features.

Currently, there is a notable lack of comprehensive datasets for the cross-view object geo-localization task. The only existing dataset is the CVOGL dataset, introduced by Sun et al. (2023), which consists of two sub-datasets: one designed for the "Ground→Satellite" localization task and the other for "Drone→Satellite" localization task. To expand the scope of cross-view object localization, we developed a new dataset specifically for the "Ground→Drone" localization task, named G2D. This dataset addresses a critical gap in the field by extending geo-localization to include interactions between ground and drone perspectives, enriching the available resources for this challenging problem. Our contributions can be summarized as follows:

1) We propose a Cross-view and Cross-attention Module (CVCAM) that leverages cross-attention to associate contextual information about the query object across different views, establishing implicit correspondences while suppressing irrelevant edge noise.

2) We introduce a Multi-head Spatial Attention Module (MHSAM) that utilizes convolutional kernels of varying sizes to capture multi-scale spatial information, further enhancing the feature representation of the query object.

3) We create the G2D dataset, enriching the data resources for cross-view object geo-localization and filling the gap in the "Ground→Drone" localization task.

4) Extensive experiments on the CVOGL and G2D datasets demonstrate that our proposed method, AttenGeo, achieves superior localization accuracy, significantly surpassing the state-of-the-art.

## 2 RELATED WORK

### 2.1 OBJECT-BASED CROSS-VIEW GEO-LOCALIZATION

Existing methods for object geo-localization aim to develop algorithms that merge repeated detections of each object into a single prediction for each ground truth object. Chaabane et al. (2021)

and Wilson et al. (2022) implemented a tracker-based approach that recognizes and merges the same object across multiple frames. Krylov et al. (2018) employed a triangulation-based method to find and merge nearby objects using triangulation algorithms. Nassar et al. (2019) utilized a re-identification approach, employing an object detector that implicitly merges repeated detections from input frames by receiving multiple frames as input. Although these methods are effective, they primarily focus on localizing the same object from similar viewpoints and neglect the localization of the same object under significantly different viewpoints. Sun et al. (2023) were the first to propose a cross-view object geo-localization task, aiming to locate objects under significant viewpoint differences. They created the CVOGL dataset for this task and introduced the DetGeo method to determine the position of query objects in reference images. DetGeo (Sun et al., 2023) uses a dual-stream CNN architecture and incorporates a query-aware cross-view fusion module—an attention mechanism to integrate feature maps from two different views to find the spatial correspondence of query objects.

## 2.2 TRANSFORMER AND ATTENTION MECHANISM

The Transformer model, first introduced by Vaswani et al. (2017), has revolutionized natural language processing (NLP) through its innovative use of the cross-attention mechanism. This mechanism establishes dependencies between sequences of different lengths, making it particularly effective for sequence-to-sequence tasks. Given the remarkable success of Transformers, attention mechanisms have been widely adopted across fields such as NLP, image classification, and object detection. For example, Hu et al. (2018a) introduced the Squeeze-and-Excitation (SE) module, which focuses on recalibrating channel-wise feature responses, thereby implementing a channel-level attention mechanism. Woo et al. (2018) proposed the Convolutional Block Attention Module (CBAM), which sequentially infers attention maps for spatial and channel dimensions. More recently, attention mechanisms have been combined with multi-scale feature extraction, enabling models to better handle objects of various sizes. A prime example is the Feature Pyramid Network (Lin et al., 2017), which constructs feature pyramids to achieve superior performance in object detection. In this context, for cross-view object geo-localization, we propose the use of a Cross-view and Cross-attention Module to establish spatial correspondences of the query object across different views, followed by the introduction of a Multi-head Spatial Attention Module to enhance the feature representation of the query object.

## 3 METHODOLOGY

### 3.1 PRELIMINARIES

**Problem Formulation.** For a given query image $q$ with the query object position $p$, our goal is to localize the position $b$ of this object in the reference image $r$. Here, the query object position $p$ in the query image is represented by the coordinate point $(x_q, y_q)$, and the query object position $b$ in the reference image is represented by the bounding box $(x_r, y_r, w, h)$. The problem equation can be described as:

$$\{b_k\}_{k=1}^N \leftarrow \{(p_k, q_k, r_k)\}_{k=1}^N$$

where $N$ represents the number of samples.

**Model Overview.** The architecture of our AttenGeo model is illustrated in Figure 2. The original query image $\mathbf{I}_q^{ori}$ is concatenated with its position-encoded (refer to Sun et al. (2023)) coordinate representation to obtain concatenated features $\mathbf{I}_q^{cat}$. The concatenated features $\mathbf{I}_q^{cat}$ are then processed through a feature extraction step to produce the feature representation $\mathbf{F}_q \in \mathbb{R}^{C \times H_q \times W_q}$. The original reference image $\mathbf{I}_r^{ori}$ is directly fed into a convolutional network to obtain the feature representation $\mathbf{F}_r \in \mathbb{R}^{C \times H_r \times W_r}$. $\mathbf{F}_q$ and $\mathbf{F}_r$ are then fused using the cross-view and cross-attention module, resulting in the fused feature $\mathbf{F}_{fused} \in \mathbb{R}^{C \times H_r \times W_r}$. Subsequently, a multi-head spatial attention module processes the fused features to yield the output features $\mathbf{O}_{fused} \in \mathbb{R}^{C \times H_r \times W_r}$. Similar to Redmon & Farhadi (2018), we employ an anchor-based approach to locate objects on the output feature map, with the object position ultimately represented by a bounding box.

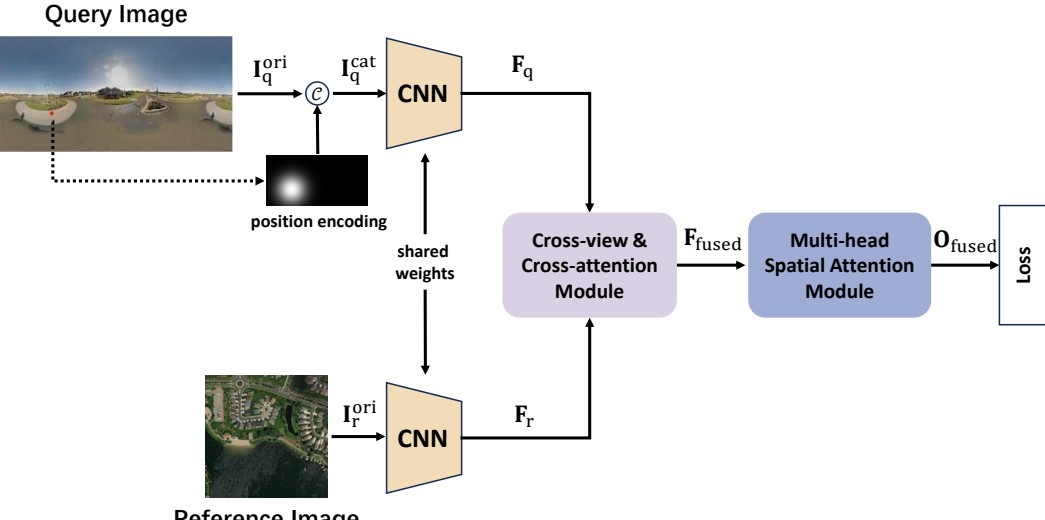

Figure 2: Overview of our proposed AttenGeo model

## 3.2 CROSS-VIEW AND CROSS-ATTENTION MODULE

This section begins with outlining the feature fusion process and then elaborates on the key components of the Cross-view and Cross-attention Module (CVCAM). The structure of CVCAM is shown in Figure 3(a). Specifically, features from the query image $\mathbf{F}_q$ and the reference image $\mathbf{F}_r$ are flattened and passed through a series of two CABs for $k$ iterations of feature interaction. This process produces the updated feature representations $\mathbf{F}'_q \in \mathbb{R}^{C \times H_q W_q}$ and $\mathbf{F}'_r \in \mathbb{R}^{C \times H_r W_r}$. Following this interaction stage, we employ an additional CAB to merge features from the two views, yielding the fused representation $\mathbf{F}_{fused} \in \mathbb{R}^{C \times H_r \times W_r}$.

**Multi-head Attention.** The key component of the Transformer is the multi-head attention block. The input feature representations are converted into three linear projections with dimension $d$, conventionally named query ($\mathbf{Q}$), key ($\mathbf{K}$) and value ($\mathbf{V}$), as the input of the attention layers. The attention operation is denoted as:

$$\text{Attention}(\mathbf{Q}, \mathbf{K}, \mathbf{V}) = \text{softmax}\left(\frac{\mathbf{Q}\mathbf{K}^T}{\sqrt{d}}\right)\mathbf{V} \tag{1}$$

Intuitively, the query $\mathbf{Q}$ retrieves information from the value $\mathbf{V}$ based on the attention weight. which is computed from the scale dot-product of $\mathbf{Q}$ and the key $\mathbf{K}$ corresponding to each value $\mathbf{V}$. In practice, each $\mathbf{Q}$, $\mathbf{K}$ and $\mathbf{V}$ is typically divided into $m$ different heads, and all the $m$ heads participate in attention operation in parallel, as these multiple heads consider information from different representation subspaces at different positions.

**Positional Encoding.** Positional encoding is added to retain spatial information. Following DETR (Carion et al., 2020), we apply fixed 2D positional encodings generated by sine functions:

$$\text{PE}_{((x,y),4j)} = \sin\left(x/10000^{2j/d_{model}}\right), \quad \text{PE}_{((x,y),4j+1)} = \cos\left(x/10000^{2j/d_{model}}\right)$$
$$\text{PE}_{((x,y),4j+2)} = \sin\left(y/10000^{2j/d_{model}}\right), \quad \text{PE}_{((x,y),4j+3)} = \cos\left(y/10000^{2j/d_{model}}\right) \tag{2}$$

where $(x, y)$ represents the 2D position, $d_{model}$ represents the feature channel dimension, and $j$ represents the index of the feature channel. The generated positional encodings provide unique spatial information for each element in the feature vector. Unlike ViT (Dosovitskiy et al., 2021), which adds positional encodings only once at the output of the backbone network, we apply fixed positional encodings to both the query and key in each cross-attention block, as shown in Fig. 3(b).

**Cross-attention Block.** The structure of our cross-attention block is illustrated in Fig. 3(b). The cross-attention block receives feature representations from two different perspective images, $\mathbf{F}_1 \in \mathbb{R}^{D \times H_1 W_1}$ and $\mathbf{F}_2 \in \mathbb{R}^{D \times H_2 W_2}$, to generate query $\mathbf{Q} \in \mathbb{R}^{D \times H_1 W_1}$, key $\mathbf{K} \in \mathbb{R}^{D \times H_2 W_2}$, and value $\mathbf{V} \in \mathbb{R}^{D \times H_2 W_2}$:

$$\mathbf{Q} = \mathbf{W}_Q \times \mathbf{F}_1, \quad \mathbf{K} = \mathbf{W}_K \times \mathbf{F}_2, \quad \mathbf{V} = \mathbf{W}_V \times \mathbf{F}_2 \tag{3}$$

where $\mathbf{W}_Q$, $\mathbf{W}_K$, and $\mathbf{W}_V$ are the learnable weights, and "$\times$" denotes matrix multiplication. The attention scores $\mathbf{Scores} \in \mathbb{R}^{H_1 W_1 \times H_2 W_2}$ are then computed using the query $\mathbf{Q}$ and key $\mathbf{K}$:

$$\mathbf{Scores} = \mathrm{softmax}\left(\mathbf{Q}^{\mathrm{T}} \times \mathbf{K}\right) \tag{4}$$

Finally, the output interactive/fused feature $\mathbf{F} \in \mathbb{R}^{D \times H_1 W_1}$ is calculated using the attention scores $\mathbf{Scores}$ and the value $\mathbf{V}$:

$$\mathbf{F} = \mathrm{rearrange}\left(\mathbf{Scores} \times \mathbf{V}^{\mathrm{T}}\right) \tag{5}$$

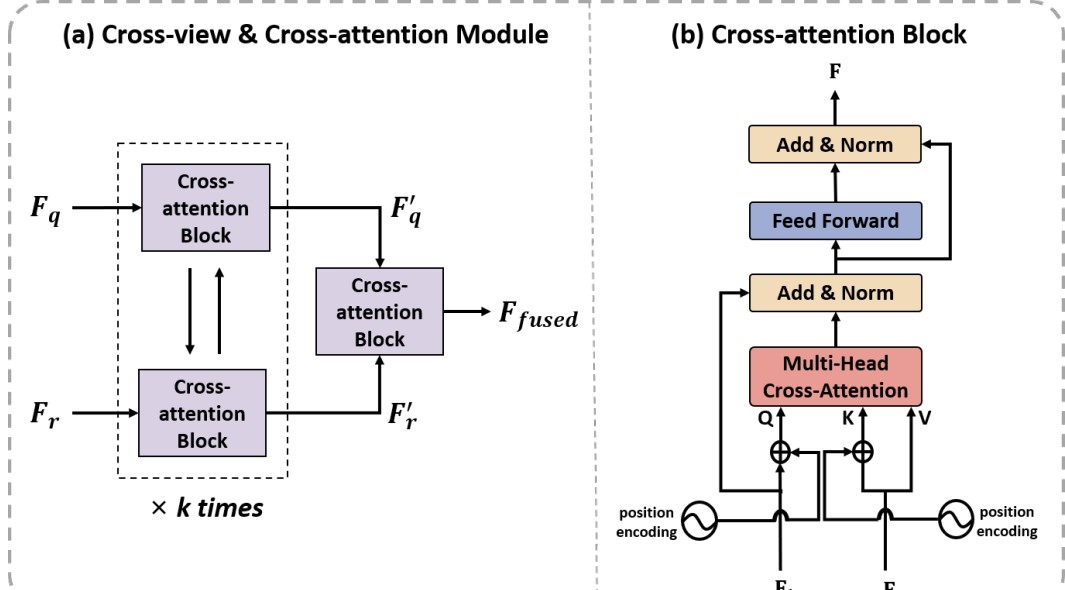

Figure 3: Cross-view and cross-attention module

## 3.3 MULTI-HEAD SPATIAL ATTENTION MODULE

The structure of our Multi-head Spatial Attention Module (MHSAM) is shown in Figure 4. MHSAM comprises three heads, each denoted as $h_i$, where $i \in 1, 2, 3$. Each attention head is composed of a convolution layer followed by a deconvolution layer. The convolution layer reduces the spatial dimensions of the input features while increasing the channel dimensions, and the subsequent deconvolution layer reverses this operation, restoring the spatial dimensions of the output tensor to match the input tensor $\mathbf{F}_{fused}$. For the $i$-th attention head $h_i$, we set the kernel size $k_i$ to be $2(i-1)+1$. In addition, we set a stride of 1 and a padding of 0 for each convolution and deconvolution layer. Overall, the output $\mathbf{H}_i$ of the $i$-th head $h_i$ is calculated as follows:

$$\mathbf{H}_i = h_i\left(\boldsymbol{\theta}_{c_i}, \boldsymbol{\theta}_{d_i}, \mathbf{F}_{fused}\right) = \max\left(0, \mathbf{F}_{fused} * \boldsymbol{\theta}_{c_i}\right) \circledast \boldsymbol{\theta}_{d_i} \tag{6}$$

where $\boldsymbol{\theta}_{c_i}$ denotes the trainable weights of the convolution layer and $\boldsymbol{\theta}_{d_i}$ denotes the trainable weights of the deconvolution layer. The operation $\max(0, \cdot)$ denotes the ReLU function, while $*$ denotes the convolution operation and $\circledast$ denotes the deconvolution operation. We observe that the outputs of the three attention heads, $\mathbf{H}_1$, $\mathbf{H}_2$, and $\mathbf{H}_3$, have the same shape. Subsequently, we sum these three tensors and apply the sigmoid function to obtain the weight tensor $\mathbf{A}$:

$$\mathbf{A} = \sigma\left(\sum_{i=1}^{3} \mathbf{H}_i\right) \tag{7}$$

where $\sigma(\cdot)$ denotes the sigmoid function. Finally, we multiply the weight tensor $\mathbf{A}$ by the module input $\mathbf{F}_{fused}$ to obtain the output tensor $\mathbf{O}_{fused}$, which incorporates spatial information with different receptive fields:

$$\mathbf{O}_{fused} = \mathbf{A} \otimes \mathbf{F}_{fused} \tag{8}$$

where $\otimes$ denotes the element-wise multiplication.

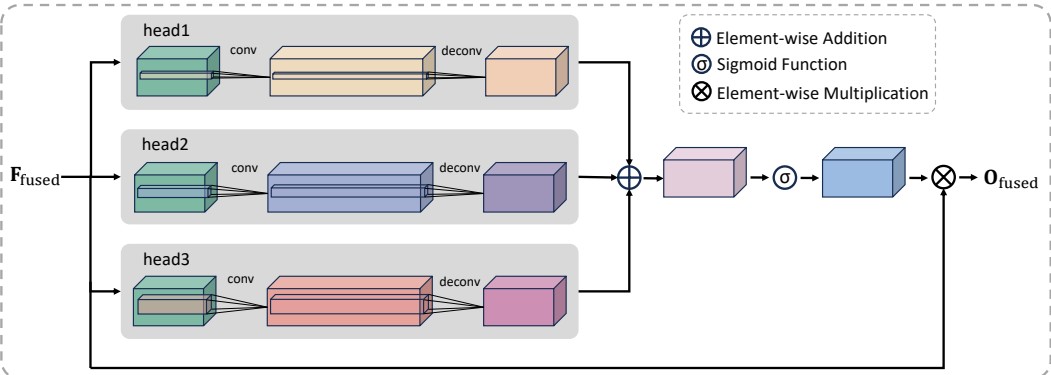

Figure 4: Multi-head spatial attention module

### 3.4 LOSS FUNCTION

The loss function in this paper is defined as follows:

$$\mathcal{L} = \mathcal{L}_{conf} + \mathcal{L}_{loc} \tag{9}$$

where $\mathcal{L}_{conf}$ denotes the confidence loss and $\mathcal{L}_{loc}$ denotes the localization loss. The confidence loss $\mathcal{L}_{conf}$ is implemented using the binary cross-entropy function (De Boer et al., 2005):

$$\mathcal{L}_{conf} = -\frac{1}{N} \sum_{k=1}^{N} (c_k \log(\hat{c}_k) + (1 - c_k) \log(1 - \hat{c}_k)) \tag{10}$$

where $\hat{c}_k$ denotes the predicted probability of the confidence for all bounding boxes of the $k$-th sample, and $c_k$ denotes the corresponding ground truth label. When the prior box corresponding to the predicted box (nine prior boxes with different aspect ratios are pre-set using clustering methods in this paper) has the maximum IoU value with the ground truth box, the corresponding $c_k$ is assigned a label of 1, while other labels are set to 0. The localization loss is implemented using mean squared error:

$$\mathcal{L}_{loc} = \frac{1}{N} \sum_{k=1}^{N} (\hat{x}_k - (x_k - \lfloor x_k \rfloor))^2 + (\hat{y}_k - (y_k - \lfloor y_k \rfloor))^2$$
$$+ (\hat{w}_k - \log(w_k/w_a))^2 + (\hat{h}_k - \log(h_k/h_a))^2 \tag{11}$$

where $\lfloor \cdot \rfloor$ denotes the floor function. $(\hat{x}_k, \hat{y}_k)$ denotes the predicted offset values for the center coordinates of the query object. $(x_k, y_k)$ denotes the center coordinates of the ground truth box. $\hat{w}_k$ and $\hat{h}_k$ denote the predicted offset values for the width and height of the query object. $w_k$ and $h_k$ denote the width and height of the ground truth box. $w_a$ and $h_a$ denote the width and height of the prior box. During the inference phase, we select the prediction box with the highest confidence score among all predicted boxes as the position prediction result for the query object:

$$\hat{b}_k = \underset{l \in \{1,2,\dots,M\}}{\arg\max} (C_{k,l}) \tag{12}$$

where $C_{k,l}$ denotes the confidence score of the $l$-th predicted box for the $k$-th sample, $M$ denotes the total number of predicted boxes, $\hat{b}_k$ denotes the predicted box output by the model, which is the position prediction result for the query object.

Table 1: Comparison with state-of-the-art methods on the CVOGL dataset. "G→S" denotes "Ground→Satellite" and "D→S" denotes "Drone→Satellite". The same notation applies in subsequent sections. Bold denotes the best results among all methods and underscore "_" denotes the second-best result.

| Method | G → S | | | | D → S | | | |
| | Test | | Validation | | Test | | Validation | |
| | acc@ 0.25(%) | acc@ 0.5(%) | acc@ 0.25(%) | acc@ 0.5(%) | acc@ 0.25(%) | acc@ 0.5(%) | acc@ 0.25(%) | acc@ 0.5(%) |
| --- | --- | --- | --- | --- | --- | --- | --- | --- |
| CVM-Net (Hu et al., 2018b) | 4.73 | 0.51 | 5.09 | 0.87 | 20.14 | 3.29 | 20.04 | 3.47 |
| RK-Net (Lin et al., 2022) | 7.40 | 0.82 | 8.67 | 0.98 | 19.22 | 2.67 | 19.94 | 3.03 |
| L2LTR (Yang et al., 2021) | 10.69 | 2.16 | 12.24 | 1.84 | 38.95 | 6.27 | 38.68 | 5.96 |
| Polar-SAFA (Shi et al., 2019) | 20.66 | 3.19 | 19.18 | 2.71 | 37.41 | 6.58 | 36.19 | 6.39 |
| TransGeo (Zhu et al., 2022) | 21.17 | 2.88 | 21.67 | 3.25 | 35.05 | 6.37 | 34.78 | 5.42 |
| SAFA (Shi et al., 2019) | 22.20 | 3.08 | 20.59 | 3.25 | 37.41 | 6.58 | 36.19 | 6.39 |
| DetGeo (Sun et al., 2023) | 45.43 | 42.24 | 46.70 | 43.99 | 61.97 | 57.66 | 59.81 | 55.15 |
| **AttenGeo(Ours)** | **50.57** | **46.15** | **49.19** | **44.10** | **70.71** | **62.08** | **68.69** | **61.86** |

## 4 EXPERIMENT

### 4.1 DATASET AND EVALUATION METRICS

**CVOGL dataset.** CVOGL (Sun et al., 2023) dataset can be used to study two types of cross-view object geo-localization tasks: Task 1 (Ground→Satellite) where ground-view images serve as query images and satellite images serve as reference images, and Task 2 (Drone→Satellite) where drone-view images serve as query images and satellite images serve as reference images. The CVOGL (Sun et al., 2023) dataset comprises a total of 12,478 sample pairs, with each sub-dataset containing 6,239 pairs. Specifically, each sub-dataset is divided into 4,343 pairs for training, 923 pairs for validation, and 973 pairs for testing.

**G2D dataset.** Our G2D dataset is primarily designed for studying the "Ground→Drone" geo-localization task, where ground-view images serve as query images and drone-view images are used as reference images. The G2D dataset contains a total of 2,753 sample pairs, with 1,951 pairs designated for training, 432 pairs for validation, and 370 pairs for testing.

**Evaluation metrics.** In line with Sun et al. (2023), we use accu@0.25 and accu@0.5 as evaluation metrics for all methods. (please refer to the appendix for more details on the evaluation metrics)

### 4.2 IMPLEMENTATION DETAILS

We employ the pre-trained ConvNeXt V2-Tiny (Woo et al., 2023) as the backbone for both the query and reference image branches, sharing weights across the dual-branch architecture. The training process utilizes the AdamW optimizer (Loshchilov & Hutter, 2019), with an initial learning rate of 0.0001, which is decayed every 10 epochs. Training is conducted over 30 epochs using two 80GB NVIDIA A100 GPUs.

### 4.3 COMPARISON WITH SATE-OF-THE-ART METHODS

We compared our method with state-of-the-art methods on CVOGL and G2D datasets. The experimental results are presented in Table 1 and Table 2. Note that the seventh model, DetGeo (Sun et al., 2023), is the first model specifically designed for the cross-view object geo-localization and the primary benchmark model for our evaluations. The experimental results demonstrate that our method surpasses all other methods on both the CVOGL and G2D datasets, proving the superiority and effectiveness of our proposed method.

Table 2: Comparison with state-of-the-art methods on the G2D dataset. "G→D" denotes "Ground→Drone". The same notation applies in subsequent sections. Bold denotes the best results among all methods and underscore "_" denotes the second-best result.

| Method | G → D | | | |
| | Test | | Validation | |
| | acc@0.25(%) | acc@0.5(%) | acc@0.25(%) | acc@0.5(%) |
|---|---|---|---|---|
| CVM-Net (Hu et al., 2018b) | 4.97 | 1.08 | 5.08 | 1.23 |
| RK-Net (Lin et al., 2022) | 13.84 | 1.47 | 12.85 | 1.58 |
| L2LTR (Yang et al., 2021) | 19.35 | 3.81 | 18.57 | 3.61 |
| Polar-SAFA (Shi et al., 2019) | 16.64 | 3.15 | 15.65 | 2.86 |
| TransGeo (Zhu et al., 2022) | 20.02 | 1.48 | 21.12 | 4.23 |
| SAFA (Shi et al., 2019) | 16.83 | 3.05 | 14.89 | 2.31 |
| DetGeo (Sun et al., 2023) | 77.03 | 72.97 | 72.69 | 68.98 |
| **AttenGeo(Ours)** | **77.30** | **74.59** | **75.00** | **70.60** |

Table 3: The ablation study of our CVCAM and MHSAM on the CVOGL and G2D datasets. "baseline" refers to our model without the CVCAM and MHSAM components.

| Dataset | Task | Method | Test | | Validation | |
| | | | acc@ 0.25(%) | acc@ 0.5(%) | acc@ 0.25(%) | acc@ 0.5(%) |
|---|---|---|---|---|---|---|
| CVOGL | G → S | baseline | 44.56 | 39.93 | 43.89 | 39.01 |
| | | baseline+CVCAM | 47.58 | 40.08 | 46.80 | 39.65 |
| | | baseline+CVCAM+MHSAM(Ours) | **50.57** | **46.15** | **49.19** | **44.10** |
| | D → S | baseline | 49.43 | 44.19 | 47.78 | 42.25 |
| | | baseline+CVCAM | 64.44 | 53.03 | 62.62 | 52.00 |
| | | baseline+CVCAM+MHSAM(Ours) | **70.71** | **62.08** | **68.69** | **61.86** |
| G2D | G → D | baseline | 74.59 | 68.92 | 73.84 | 68.75 |
| | | baseline+CVCAM | 75.95 | 70.00 | 74.07 | 69.21 |
| | | baseline+CVCAM+MHSAM(Ours) | **77.30** | **74.59** | **75.00** | **70.60** |

## 4.4 ABLATION ANALYSIS

**Effectiveness of main components.** We conducted an ablation study by removing the CVCAM and MHSAM components from our AttenGeo model, using a simple summation of the features from both views as our baseline. Based on this baseline, we evaluated the effectiveness of integrating our proposed CVCAM and MHSAM components. The experimental results are shown in Table 3. The results demonstrate that with the successive addition of CVCAM and MHSAM, the model's performance improves incrementally. This performance boost is particularly evident in the D→S geo-localization task, confirming the effectiveness of our CVCAM and MHSAM.

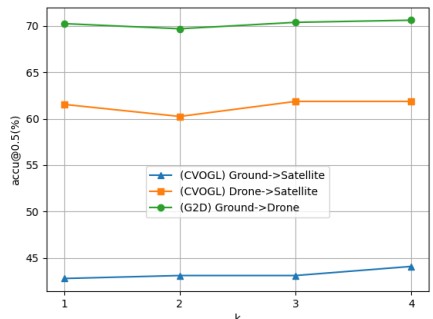

Figure 5: Accu@0.5(%) for varying $k$ on the validation sets of CVOGL and G2D datasets.

**Evaluation of hyperparameter $k$.** We evaluated the effect of varying interaction counts $k$ in the CV-

CAM on model performance using the validation
sets of the CVOGL and G2D datasets, as shown in Figure 5. The results indicate that when $k$
is set to 4, our AttenGeo achieves improved performance on both the CVOGL and G2D datasets.
Therefore, we set the value of $k$ to 4 for the CVCAM in this study.

## 4.5 VISUALIZATION ANALYSIS

To further evaluate the effectiveness of our CVCAM and MHSAM, we conducted a visualization
analysis of the output feature maps from both modules and compared them with the Query-aware
Cross-view Fusion Module (QACVFM) proposed by Sun et al. (2023), as shown in Figure 6. The
visualization in Figure 6 demonstrates that, after integrating the CVCAM module, our model is able
to capture the spatial location of the query object while suppressing the edge noise, indicating that
the model has preliminarily established an implicit correspondence between the query object across
the two views. Moreover, after further incorporating the MHSAM module, our model not only
accurately localizes the query object but also successfully suppresses interference from non-target
regions. In contrast, although DetGeo (Sun et al., 2023) responds to the query object region by
applying the QACVFM, it incorrectly attends to some edge noise. These results demonstrate that
our CVCAM and MHSAM are highly effective in enhancing the model discriminative ability and
reducing false positives. (please refer to the appendix for more visualization results)

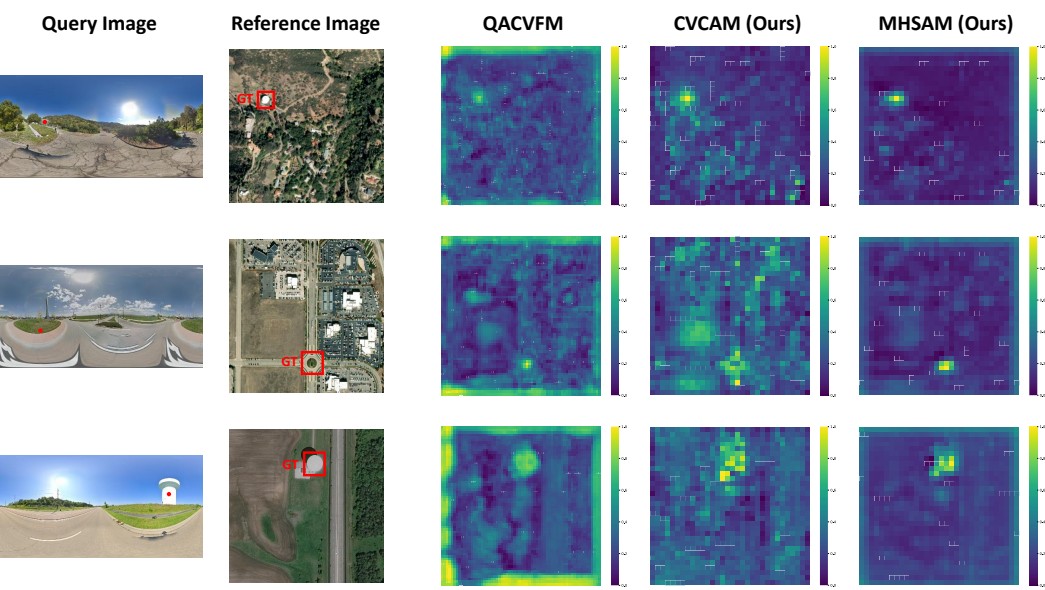

Figure 6: Visualization comparison of the output feature maps of the QACVFM (Sun et al., 2023),
CVCAM (Ours), and MHSAM (Ours).

## 4.6 PORTABILITY ANALYSIS

We integrate our CVCAM and MHSAM into DetGeo (Sun et al., 2023) to evaluate the portabil-
ity of these two modules on the CVOGL and G2D datasets. The experimental results are shown
in Table 4. The results indicate that after incorporating our MHSAM, DetGeo (Sun et al., 2023)
achieves better object localization performance on both datasets, demonstrating the good portabil-
ity of our MHSAM. However, after replacing the original QACVFM module in DetGeo with our
CVCAM, the model shows improved performance on the "G→S" task of the CVOGL dataset and
"G→D" task of the G2D dataset, but exhibits a decline in performance on the "D→S" task of the
CVOGL dataset. Based on the visualization results in Figure 6, we hypothesize that this may be
because, although CVCAM can focus on the query object region and suppress the edge noise, it also
attends to some non-target regions, which could affect the model's object localization performance
to some extent. Therefore, our CVCAM does not exhibit consistent improvements across all object

Table 4: Evaluate the portability of our CVCAM and MHSAM on the CVOGL and G2D datasets. "w/ CVCAM" refers to replacing the QACVFM with our CVCAM in DetGeo. "w/ MHSAM" refers to the incorporation of our MHSAM into DetGeo.

| Dataset | Task | Method | Test | | Validation | |
|---|---|---|---|---|---|---|
| | | | acc@ 0.25(%) | acc@ 0.5(%) | acc@ 0.25(%) | acc@ 0.5(%) |
| CVOGL | G → S | DetGeo (Sun et al., 2023) | 45.43 | 42.24 | 46.70 | 43.99 |
| | | w/ CVCAM | 48.92(+3.49) | 44.91(+2.67) | 47.56(+0.86) | 44.01(+0.02) |
| | | w/ MHSAM | 46.76(+1.33) | 43.58(+1.34) | 47.83(+1.13) | 44.58(+0.59) |
| | D → S | DetGeo (Sun et al., 2023) | 61.97 | 57.66 | 59.81 | 55.15 |
| | | w/ CVCAM | 56.22(-5.75) | 50.98(-6.68) | 53.95(-5.86) | 50.16(-4.99) |
| | | w/ MHSAM | 62.12(+0.15) | 57.98(+0.32) | 60.94(+1.13) | 55.63(+0.48) |
| G2D | G → D | DetGeo (Sun et al., 2023) | 77.03 | 72.97 | 72.69 | 68.98 |
| | | w/ CVCAM | 77.03(+0.00) | 73.78(+0.81) | 74.54(+1.85) | 69.75(+0.77) |
| | | w/ MHSAM | 78.65(+1.62) | 75.14(+2.17) | 74.54(+1.85) | 69.91(+0.93) |

localization tasks. Nevertheless, we consider that our CVCAM remains competitive compared to QACVFM (Sun et al., 2023).

## 4.7 MODEL ANALYSIS

We present an evaluation of our model performance on the test set, detailing its computational cost, parameter count, inference time, and accuracy at the accu@0.5 and comparing with DetGeo (Sun et al., 2023), as shown in Table 5. The results indicate that our model achieves a lower computational cost, fewer parameters and higher object localization accuracy, while maintaining an inference speed comparable to DetGeo (Sun et al., 2023). This demonstrates that our model outperforms DetGeo (Sun et al., 2023) overall.

Table 5: Overall performance analysis of our AttenGeo

| Dataset | Task | Method | GFLOPs | Params (M) | Inference time(ms) | accu@ 0.5(%) |
|---|---|---|---|---|---|---|
| CVOGL | G→S | DetGeo (Sun et al., 2023) | 206.94 | 73.80 | **42.04** | 42.24 |
| | | AttenGeo(Ours) | **116.43** | **42.15** | 62.34 | **46.15** |
| | D→S | DetGeo (Sun et al., 2023) | 204.54 | 73.80 | **39.77** | 57.66 |
| | | AttenGeo(Ours) | **110.46** | **42.15** | 58.13 | **62.08** |
| G2D | G→D | DetGeo (Sun et al., 2023) | 206.94 | 73.80 | **41.64** | 72.97 |
| | | AttenGeo(Ours) | **116.43** | **42.15** | 61.14 | **74.59** |

## 5 CONCLUSION

In this paper, we propose the CVCAM to enable the model to fully learn the contextual information of the query object across different views, establishing implicit correspondences while suppressing edge noise. Furthermore, we introduce the MHSAM to enhance the feature representation of the query object, improving the model ability to capture the local features of the query object. In addition, we create the G2D dataset, which enriches the datasets for cross-view object geo-localization and fills the gap for "Ground→Drone" localization task. Extensive experiments on the CVOGL and G2D datasets demonstrate that our method (AttenGeo) achieves superior object localization performance, reaching the state-of-the-art level.

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

# APPENDIX

## A   EVALUATION METRICS

In line with Sun et al. (2023), we adopt the accuracy metric accu@t calculated using Intersection over Union (IoU) as the evaluation metric. The definition of accu@t is as follows:

$$accu@t = \frac{1}{N}\sum_{k=1}^{N}\varphi(k), \ \varphi(k) = \begin{cases} 1, & \text{if } \text{IoU}\left(\hat{b}_k, b_k\right) \geq t \\ 0, & \text{else} \end{cases}, \ \text{IoU}\left(\hat{b}_k, b_k\right) = \frac{\left|\hat{b}_k \cap b_k\right|}{\left|\hat{b}_k \cup b_k\right|} \quad (13)$$

where $\hat{b}_k$ represents the predicted box for the $k$-th sample, $b_k$ represents the ground truth box for the $k$-th sample, $\left|\hat{b}_k \cap b_k\right|$ represents the intersection area between $\hat{b}_k$ and $b_k$, and $\left|\hat{b}_k \cup b_k\right|$ represents the union area between $\hat{b}_k$ and $b_k$. $t$ represents the threshold. We consider the prediction to be correct when the IoU is greater than or equal to the threshold. We use accu@0.25 and accu@0.5 as evaluation metrics for all methods.

## B   MORE VISUALIZATION ANALYSIS

We present a heatmap visualization of the object localization results obtained by our AttenGeo model, as shown in Figure 7. The results demonstrate that our AttenGeo effectively focuses on the query object within the image, achieving precise localization of its spatial position. This observation underscores the practical efficacy of our proposed method, highlighting its capability to enhance object localization performance.

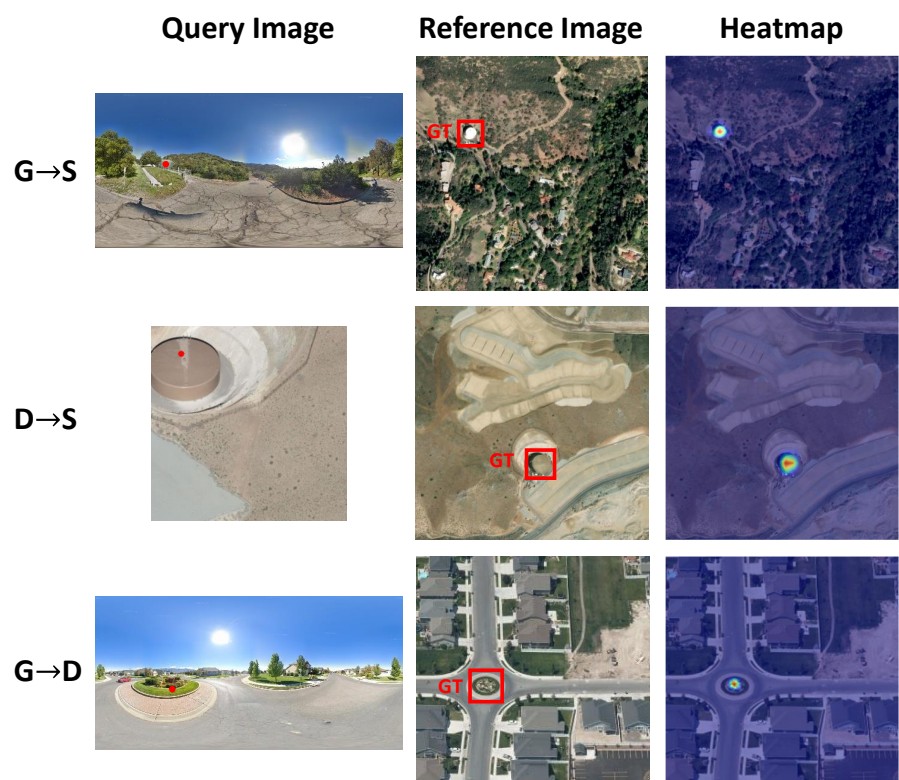

Figure 7: Heatmap visualization of the localization results for the query object by our AttenGeo.

