# OpenReview forum: "Improving Cross-view Object Geo-localization: A Dual Attention Approach with Cross-view Interaction and Multi-Scale Spatial Features"
_ICLR.cc/2025/Conference — ICLR 2025 Conference Withdrawn Submission_

### Official Review · Reviewer_d7xG · 2024-10-31

**Soundness:** 2
**Presentation:** 3
**Contribution:** 2
**Rating:** 5
**Confidence:** 4

**Summary:**

Current geo-localization through attention mechanisms focus more on irrelevant edge noise. To address the limitations, the paper proposed to iteratively refine spatial relationship feature maps. Meanwhile, the authors integrate a multi-head spatial attention module. The paper presents incremental improvements over existing methods and contributes a new dataset to the field. While the technical novelty is limited, the empirical results and new dataset make it a useful contribution.

**Strengths:**

The paper is generally well-written and organized. The experimental results seems strong.

**Weaknesses:**

- The core components (CVCAM and MHSAM) are largely based on existing techniques like Transformer's cross-attention and basic convolution operations. It is better to Clearer explanation of technical innovations beyond existing methods with theoretical analysis of why the proposed approach works better.

- How to justify the motivation of using multiple iterations of cross attention? How sensitive is the performance to the architecture choices?

- Analyze what types of cases show improvement vs. where the method fails.

- While one contribution of this paper is G2D dataset, however the descriptions of the dataset is too few. Such as, dataset collection and annotation process, comparison with existing datasets and potential biases or limitations etc.

**Questions:**

Please refer to weaknesses part for more details on the questions.

---

### Official Review · Reviewer_cxCX · 2024-11-02

**Soundness:** 2
**Presentation:** 3
**Contribution:** 2
**Rating:** 3
**Confidence:** 3

**Summary:**

The paper aims to improve object localization across views by leveraging a Cross-View and Cross-Attention Module (CVCAM) for better feature interaction and a Multi-Head Spatial Attention Module (MHSAM) for enhanced spatial representation. The authors also present G2D, a new dataset for Ground-to-Drone localization. The experimental results show that the proposed method achieves state-of-the-art results, setting a new benchmark for cross-view geo-localization.

**Strengths:**

1. The paper is well-structured and organized, presenting technical content in an accessible, easy-to-follow manner. Each section flows logically, allowing readers to understand the proposed approach's motivation and functionality.

2. Introducing the G2D dataset for Ground-to-Drone localization addresses a notable gap in existing resources, enabling new research into localization tasks across ground and aerial views. This dataset is a valuable addition, supporting further exploration and benchmarking in this challenging domain.

3. Experimental results demonstrate the effectiveness of the proposed method, with consistent improvements across key performance metrics. These results validate the model's enhancements and highlight its potential for advancing the field of cross-view object geo-localization.

**Weaknesses:**

1. The paper leans heavily towards an engineering approach, with its primary contributions being the two attention-based modules, CVCAM and MHSAM, which resemble components commonly seen in existing research. While these modules are well-integrated, the paper lacks a deeper level of novelty, as it does not provide mathematical proofs or theoretical insights to support the unique effectiveness of these modules.

2. The evaluation is limited in scope, as it relies on only two datasets (CVOGL and G2D) for benchmarking. A broader set of experiments across multiple datasets or real-world scenarios would enhance the generalizability and robustness of the results and demonstrate the method's effectiveness under varied conditions.

3. The paper lacks sufficient analysis on why the proposed modules—particularly CVCAM and MHSAM—improve cross-view localization. While some visualization results are provided, a more detailed analysis of how each module specifically impacts performance, supported by thorough visualizations, would help clarify their contributions and effectiveness in the model’s performance.

**Questions:**

please see the Weaknesses

---

### Official Review · Reviewer_C5NW · 2024-11-02

**Soundness:** 2
**Presentation:** 2
**Contribution:** 2
**Rating:** 3
**Confidence:** 4

**Summary:**

This paper proposes a network for cross-view object geo-localization, composed of a Cross-View and Cross-Attention Module (CVCAM) and a Multi-Head Spatial Attention Module (MHSAM). Additionally, the authors introduce a new dataset, named G2D, specifically designed for cross-view object geo-localization. Experimental results show that their method, AttenGeo, outperforms state-of-the-art techniques on both the CVOGL and G2D datasets.

**Strengths:**

1.The experiments demonstrate that AttenGeo achieves a high level of localization accuracy, significantly surpassing state-of-the-art methods, which suggests the effectiveness of the proposed approach.
2.The introduction of the G2D dataset is a valuable contribution to the cross-view object geo-localization research community, and it can facilitate further developments in the field.

**Weaknesses:**

1. The explanation of the model’s architecture could be clearer. For instance, in Section 3.2, the authors mention that the query and reference features are passed through two cross-attention blocks. However, only images from a single view are fed into this part of the model, why it is referred to as cross-attention rather than self-attention?
2. The novelty of this work appears limited. The use of cross-view attention and the implementation of cross-attention and spatial attention modules are widely established in the field, with several existing works using similar techniques.
3. Although the paper introduces the G2D dataset, details about its collection methodology, sensor types, and visualizations of the data are lacking. Providing these additional dataset details would enhance the paper’s transparency and help other researchers understand and utilize the dataset effectively.

**Questions:**

Please refer to weaknesses.

---

### Official Review · Reviewer_hbzD · 2024-11-03

**Soundness:** 2
**Presentation:** 2
**Contribution:** 3
**Rating:** 5
**Confidence:** 4

**Summary:**

The article demonstrates a significant amount of work and is well-structured, with clear expression. It highlights the issue that existing geo-localization methods are not effective for cross-view object localization and primarily designs two modules, CVCAM and MHSAM, to address this problem. Additionally, it creates a new dataset, G2D, to tackle the lack of cross-view object data in existing datasets for the "Ground-Drone" localization task. The proposed new method performs well on both the existing and new datasets, adapting to G2S, D2S localization tasks, and the G2D task, showcasing the generalization capability of the algorithm.

**Strengths:**

The issue raised in the article is critical, as the focus on edge noise is unrelated to localization, and the G2D data is crucial for achieving accurate and applicable geo-localization. The overall style of the article is simple and clear, making it very accessible for readers who are unfamiliar with geo-localization. The proposed algorithm achieves state-of-the-art accuracy on two cross-view datasets and demonstrates strong adaptability across different perspectives, including G2S, D2G, and G2D, showcasing its robust generalization capability.

**Weaknesses:**

Sections 3.1 and 3.2 of the article focus on the implementation details and operational processes of the two main algorithm modules, specifically regarding commonly used multi-head attention and positional encoding modules. However, they do not explain how CVCAM effectively reduces the attention on irrelevant edge noise, as mentioned in the introduction, nor do they clarify how MHSAM further processes the spatial relationships. The principles behind how these two modules address the identified problems are not detailed, which diminishes their persuasive power. Since the technological foundations of both modules are relatively common, failing to connect them to the cross-view geo-localization task and explain how they solve the proposed issues may lead to a perception of insufficient innovation.

**Questions:**

1.	How does the CVCAM module proposed in the article address the issue of the network overly focusing on edge noise through cross-attention? What is the difference between cross-attention and the dot product used in previous methods in this task?
2.	How does MHSAM achieve accurate attention focus on the target object through multi-scale convolution and deconvolution?

---

### Official Review · Reviewer_TWre · 2024-11-04

**Soundness:** 2
**Presentation:** 3
**Contribution:** 2
**Rating:** 5
**Confidence:** 4

**Summary:**

This paper proposes the CVCAM and MHSAM models to learn the contextual information of the query object, establishing implicit correspondences while suppressing edge noise and capturing local features. It also introduces the G2D dataset, enriching resources for cross-view object geo-localization and addressing the gap in 'Ground→Drone' localization tasks.

**Strengths:**

1. The paper is well-written.
2. The method achieves state-of-the-art performance with lower computational cost.

**Weaknesses:**

1. The method employs a cross-attention transformer (CVCAM) to extract a fused feature map from two views, followed by a UNet (MHSAM) for capturing local information. This approach lacks novelty.

2. The dataset contribution is unclear. It seems the images are sourced from the original CVOGL dataset. Have you merely created pair lists, or have you added any additional labels to enhance the dataset?

**Questions:**

1. What is the size of weight tensor A? Is it a window size mask?

---

### Note · Authors · 2024-11-13

I have read and agree with the venue's withdrawal policy on behalf of myself and my co-authors.